# Progress on Geographical Distribution, Driving Factors and Ecological Functions of Nepalese Alder

Chenxi Xia [1,5], Wanglin Zhao [1,2], Jinniu Wang [3], Jian Sun [1], Guangshuai Cui [1] and Lin Zhang [1,2,4,*]

1   State Key Laboratory of Tibetan Plateau Earth System, Resources and Environment, Institute of Tibetan Plateau Research, Chinese Academy of Sciences, Beijing 100101, China
2   Muoto Observation and Research Center for Earth Landscape and Earth System, Institute of Tibetan Plateau Research, Chinese Academy of Sciences, Beijing 100101, China
3   Chengdu Institute of Biology, Chinese Academy of Sciences, Chengdu 610041, China
4   Tibet Autonomous Region Institute of Science and Technology Information, Lhasa 850000, China
5   University of Chinese Academy of Sciences, Beijing 100049, China
*   Correspondence: zhanglin@itpcas.ac.cn; Tel.: +86-010-8409-7055

**Abstract:** As the oldest species of Betulaceae, Nepalese alder (*Alnus nepalensis*) shows a high capacity for nitrogen fixation, rapid growth rate, and strong adaptability to stress environments, and it plays an important role in maintaining the structure and function of forest and agroforestry ecosystems. We explored its geographic distribution and the corresponding environmental drivers through collecting specimen records and published literature for Nepalese alder over the world during the past 40 years. The research trends, the growth limiting factors, the physiological characteristics, and ecological functions were all summarized as well. In terms of geographical distribution and limiting factors, Nepalese alder is mainly distributed in southern mountainous areas of the Himalayas and southwest China. Since it presented a clear northern limit of distribution and an upper limit of elevation, temperature is assumed to be the main environmental limiting factor. According to historical development, the research history of Nepalese alder could be divided into three main periods: the initial development (before 2001), the fast development (2002–2015), and the high-quality development (2016–2022), with the two key points in 2002 and 2015 relating to the conversion of cropland to a forest project that the government conducted and the application from theory to practice, respectively. As can be seen from the ecological functions, Nepalese alder could form symbiotic nodules with Frankia, which plays an important role in improving soil physical and chemical properties and facilitating vegetation secondary succession. Overall, the present review provides a reference for further studies on ecological adaptability and sustainable utilization of Nepalese alder under climate change, and also for regional ecosystem service, forestry production practice, and vegetation restoration.

**Keywords:** biological nitrogen fixation; bibliometrics; climate change; Frankia; vegetation restoration

## 1. Introduction

Nepalese alder (*Alnus nepalensis*), a plant of the alder genus *Alnus* in the family of Betulaceae, is described in the Flora of China as growing in river beach wetlands or gully terrace forests (Figure 1) at a wide range of altitudes between about 700 and 3600 m. It grows fast in warm and humid environments, with a height exceeding 13 m over 5 years [1], and it has a strong carbon sequestration capacity [2]. This species is resistant to barrenness, and is often used for riverside berms and barren mountain beautification. Thus, it has been selected as an ideal tree species for ecological shelter forest and mixed afforestation [3,4]. At the same time, as a high-quality papermaking raw material, the Nepalese alder is an important tree species used to create short-cycle industrial wood raw material forests. Therefore, this species is listed in China's "Forest Resources Development and Protection Project" (FRDPP), and it is the main broad-leaved tree species for the

construction of papermaking industries [5]. The characteristics of Nepalese alder as a pioneering species, its rapid growth, and its high-quality fuel carbon are inseparable from its nitrogen fixation performance by the symbiotic nodules where the Frankia is attached [6]. This nitrogen fixation performance increases soil organic matter and nitrogen and phosphorus nutrient content [7,8], promotes soil microbial reproduction, and increases soil enzyme activity [9]. To some extent, the improvement of soil properties also promotes the growth and development of other plants within the same area, so Nepalese alder is often introduced into agroforestry to enhance the biological production of target species. In the eastern Himalayas, agroforestry ecosystems dominated by cardamom (*Amomum subulatum*) and Nepalese alder are specially created [10], and Sharma et al. (2002b) found that the energy conversion efficiency and net energy increase of alder–cardamom were significantly higher than those of an ordinary cardamom system, and the production potential of the ecosystem was also optimized [11]. In addition, studies have pointed out that the new branches and leaves of Nepalese alder reflected high moisture content and are not prone to canopy fires. Therefore, it can be cultivated in large quantities as a biological fireproof tree species [12]. In all, the Nepalese alder plays an important role in maintaining the stability of forest ecosystems, agroforestry ecosystems, and the sustainable development of forestry.

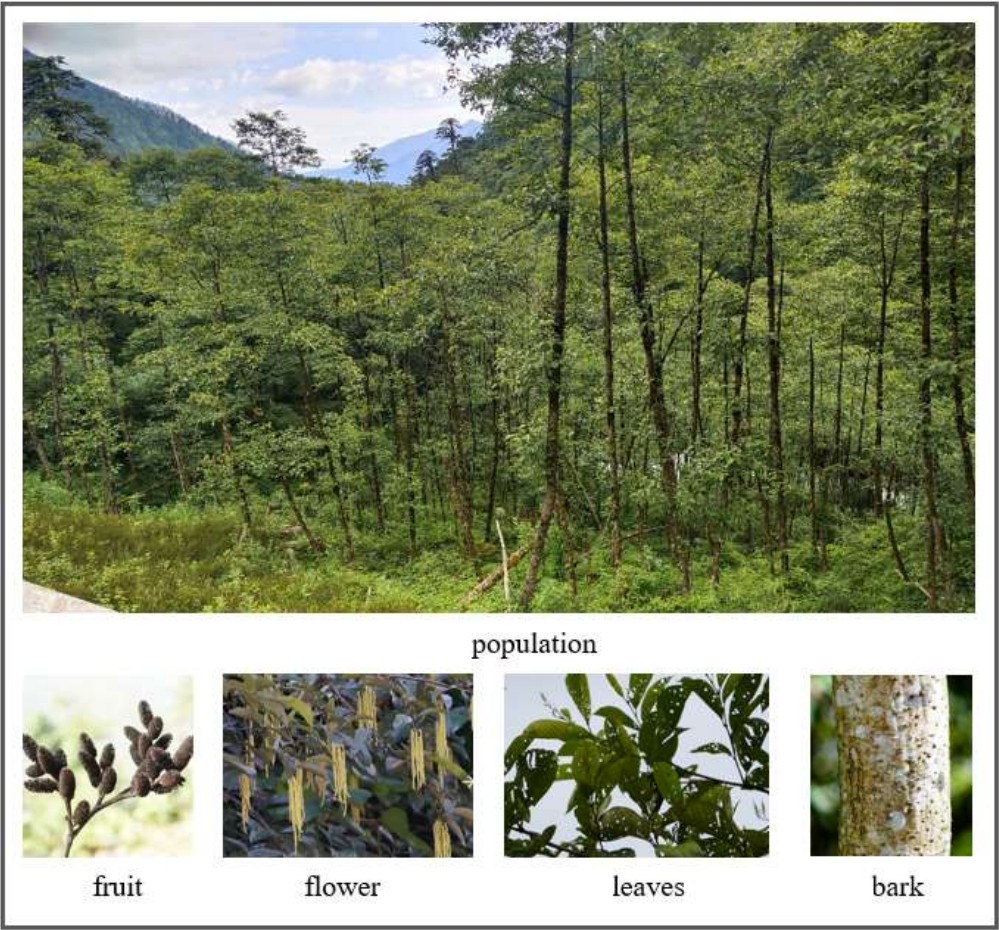

**Figure 1.** Community structure and organ characters of Nepalese alder (Motuo, Tibet).

According to the Flora of China, the distribution of Nepalese alder spans a large altitudinal gradient near 3000 m, which indicates that this species is well-adapted to different environments. In addition, the clear altitude limit (especially the upper limit) indicates that there is an obvious hydrothermal constraint. However, at present, research on the correlation between alder and environmental factors mainly focused on the effects of nitrogen- and phosphorus-addition treatments on the growth of alder seedlings in Nepal [13,14], whereas research concentrating on the relationships between alder and

hydrothermal factors such as temperature and precipitation is still lacking on a large scale [15]. In addition, Frankia attached to the roots of Nepalese alder are closely related to the nitrogen fixation ability of the tree itself, and as an actinomycete in microorganisms, its activity is also affected by environmental factors [16]. Thus, the response of Frankia to environmental factors will also affect the growth of Nepalese alder. Accordingly, this paper first discusses the global geographical distribution and limiting factors of Nepalese alder by querying and collecting the relevant literature of Nepalese alder from both domestic and foreign specimen databases, and then it analyzes the relevant research hotspots and development trends based on Citespace software and literature synthesis, respectively. We further reviewed the main physio-ecological characteristics and ecological functions of Nepalese alder, which might provide a reference for in-depth exploration of the ecological adaptability and sustainable use of alder under the background of climate change. At the same time, this paper may offer a theoretical reference for regional forestry production practice and vegetation restoration.

## 2. Data and Methods

### 2.1. Literature Collection

In the core database of the Web of Science (WOS) Science Citation Index (SCIE), a search for literature between 1981 and 2022 with a subject term of "Alnus nepalensis*" yielded a total of 140 records. In the China National Knowledge Infrastructure (CNKI) database, with the subject terms "Nepalese alder", "Handonggua", or "Mengzi alder", we found a total of 366 records. After manually removing impurities from the search data such as patents, reports, notices, and other documents, as well as some of the "Handonggua" literature which documented other alder species rather than *Alnus nepalensis*, 128 English papers and 334 Chinese documents were finally obtained.

### 2.2. Geographic Data Analysis

Based on the geographic information (longitude and latitude) of study sites in the existing literature described above (66 sites, the detailed geographic information is not available in all the papers), as well as that of Nepalese alder specimens from the Global Biodiversity Information Facility (GBIF, https://www.gbif.org, 184 sites, accessed on 30 September 2022) and the iPlant (http://www.iplant.cn, 71 sites, accessed on 30 October 2022), we analyzed the geographical distribution pattern about this species by using ArcGIS 10.7 for Desktop. The Shuttle Radar Topographic Mission Digital Elevation Model (SRTM DEM), the most comprehensive high-resolution digital topographic database, was applied to obtain the elevation for the above 321 sites. The SRTM DEM data with a spatial resolution of 90 m were obtained from the USGS (https://earthexplorer.usgs.gov/, accessed on 10 October 2022). Slope aspect of the study site was obtained by using visual interpretation of high-resolution remote sensing imagery (World Imagery, https://www.arcgis.com/, accessed on 2 November 2022).

### 2.3. Bibliometrics Analysis

With the help of CiteSpace visualization software (version number: 6.1.R3), keyword clustering analysis was carried out for the English and Chinese literature, respectively. The keyword clustering map of Nepalese alder researches were obtained, and the relevant research hotspots were revealed according to the analysis. Since the literature related to the Nepalese alder was mainly in Chinese (accounting for about 72%), we summarized the development processes about *A. nepalensis* researches based on the published papers in Chinese.

## 3. Geographic Distribution of Nepalese Alder and Corresponding Limiting Factors

As shown in Figure 2, the Nepalese alder is widely distributed, with a latitude ranging between 38°47′ S and 34°26′ N, and a longitude from 156°55′ W to 177°43′ E. It is mainly concentrated in the southern foothills of the Himalayas and the southwestern part of China

(especially Yunnan province) with a middle and low latitude of 20°01′–30°32′ N, and a longitude of 79°06′–104°20′ E. Additionally, a few of the records were found in North America, Oceania, and Ecuador in South America. Perhaps due to the synthetic influence of climatic, geological, and topographical factors, such as the southwest monsoon, the east Asia monsoon, and the uplift of the Himalayas, coupled with the shade-intolerant and adaptive characteristics of Nepalese alder, the Himalayas, Hengduan Mountain, and their southern mountains have become the main distribution area of Nepalese alder. Compared with other alder species of the congener (*A. formosana*, *A. trabeculosa*), Nepalese alder has a wider range at similar latitudes and a larger altitude span (more than 3000 m); for example, *A. formosana* is only distributed at altitudes of no more than 2500 m [17], whereas *A. trabeculosa* is only distributed between 200 and 1000 m [18]. Therefore, Nepalese alder is more adaptable to complex and varied habitats.

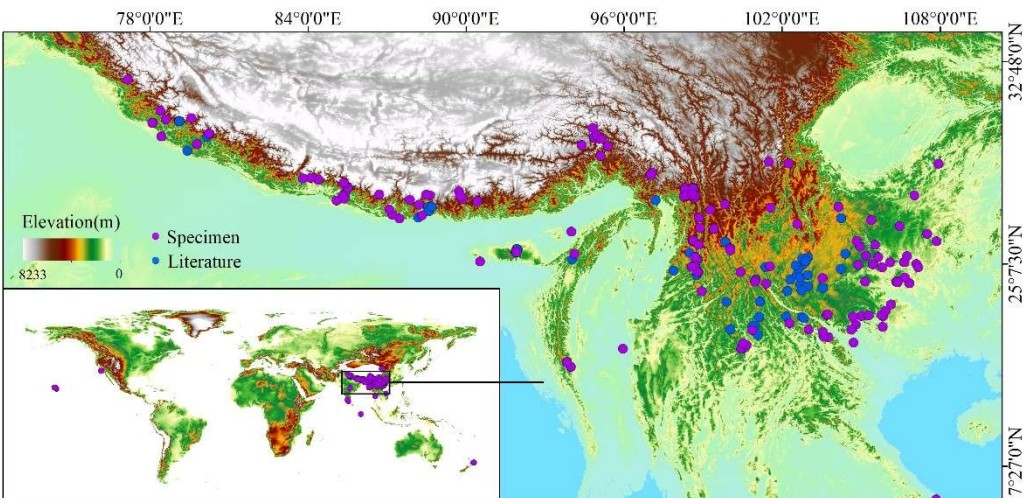

**Figure 2.** Geographic distribution of Nepalese alder based on information from specimen and literature.

From the respect of its northern boundary, the westernmost is in the district of Korlu in Himachal Pradesh in northeastern India, with a latitude of 32°60′ N. The northern boundary of the central region is mainly located in the Dongjiu area of Bayi District, Nyingchi City, with a latitude of 30°00′ N, whereas those of the eastern region were in Shimian County of Ya'an City, Sichuan Province and Wuchuan Gelao and Miao Autonomous County of Zunyi City, Guizhou Province, with latitudes of 28°56′ N and 28°55′ N, respectively. The meteorological data of the above four sites of northern limits were obtained from the WorldClim website (https://worldclim.org/, accessed on 11 October 2022). After justification according to the differences between gridded and exact altitudes, the annual mean temperatures were calculated as 15.4 °C, 9.6 °C, 10.8 °C, and 14.7 °C, respectively. Therefore, it seemed that the annual mean temperature of about 10 °C might be the thermal factor shaping its geographic distribution, which is consistent with Yang et al. who reported a low temperature limit of about 10 °C for Nepalese alder [4].

In terms of altitudinal distribution, nearly three-quarters of the research sites and specimen collection sites of Nepalese alder were concentrated between 1000 and 2500 m (Figure 3a), and only 12.79% of the sites sat below 1000 m, and a very small number (less than 1.3%) were distributed in alpine regions above 3500 m (mainly on Hengduan Mountain). The highest altitude record, 3860 m, happened in the northwest of Huili County in southern Sichuan Province, at a latitude of 26°39′ N. In addition, several sites with an altitude of more than 3600 m are distributed in Weixi Lisu Autonomous County, Yunnan, with a latitude of about 27°12′ N. Accordingly, it can be seen that the Nepalese alder is mainly distributed in typical mountainous terrain environments, and it is rarely distributed in low mountain plains and alpine areas above 3500 m.

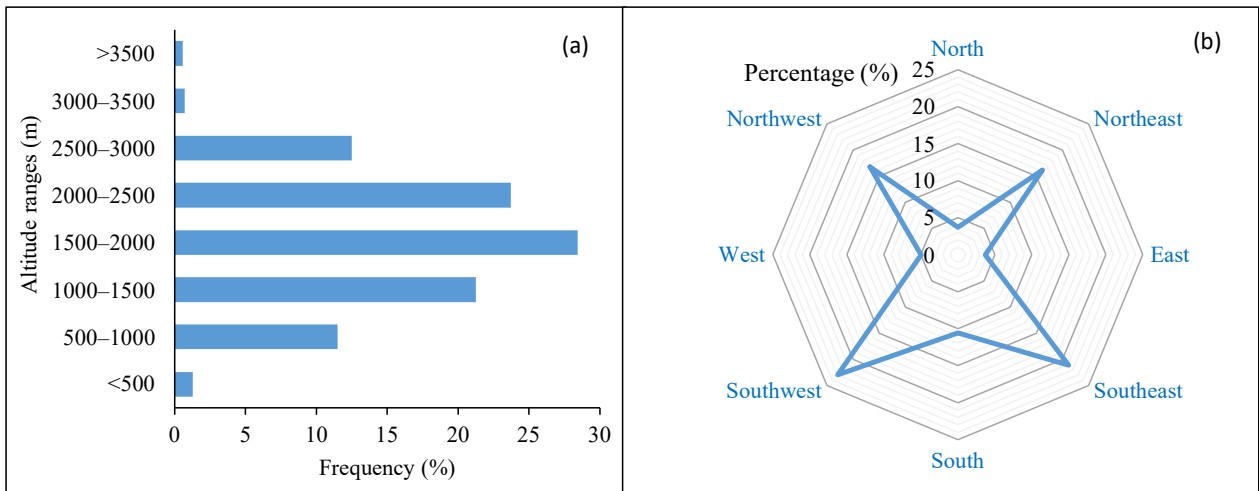

**Figure 3.** (**a**) Altitude and (**b**) slope aspect distribution patterns of Nepalese alder.

Slope aspect may also exert impacts on the distribution of Nepalese alder. Nearly 60% of the sites were distributed in south-facing slopes (Figure 3b), and only 37% of the samples were distributed in north-facing slopes, indicating that the Nepalese alder prefers sunny slope habitats, consistent with its sunward and light-loving characteristics [4]. This is also associated with its distribution in the southern foothills of the Himalayas and the mountainous areas of southwest China, both of which showed the same topographic characteristic of high in the north and low in the south.

## 4. Development History of Nepalese Alder

The whole development history can be roughly summarized into three periods: the initial development (before 2001), the fast development (2002–2015), and the high-quality development (2016–2022) (Figure 4). Among them, the number of papers published in the early period was small, with a total of only 30 papers, and the research topics mainly focused on the growth habit and forest community structure of Nepalese alder (Figure 5a). For example, the Yunnan Institute of Forestry [19] reported its growth characteristics and corresponding habit, which is a relatively systematic study of Nepalese alder in China in this stage. Guo et al. (1999) found that the alder tree layer exerted a significant effect on precipitation interception in the central Yunnan plateau [20]. In 2002, research development entered into an accelerated period, during which the number of publications increased significantly and reached a peak in 2015 (39 articles), which was about 10 times that of 2005 (the least number of articles published in this period). This might be related to the national policy to fully launch the project of returning farmland to forest in 2002, as well as the implementation of related projects which greatly promoted the research on the fast-growing alder tree species. The focus at this stage gradually shifted to vegetation restoration and seedling growth (Figure 5b). For example, Yang et al. (2004) found that the community coverage and species composition were relatively stable in phosphate mining areas [21], so it had important application value in the restoration of natural vegetation in phosphate mining wasteland. Zhou et al. (2012) found that the presence of this species significantly increased the available nitrogen and potassium content in the soil, promoted the formation of soil agglomerate structure, and effectively accelerated the restoration process of vegetation in phosphate collection areas [22]. After 2015, the number of publications showed a downward trend, which might be related to the fact that the research has focused more on practical applications. This stage focused on soil nutrients and vegetation restoration, and related research gradually shifted to practical applications (Figure 5c); for example, the Yuanyang Rice Terraces had become a research hotspot [23,24].

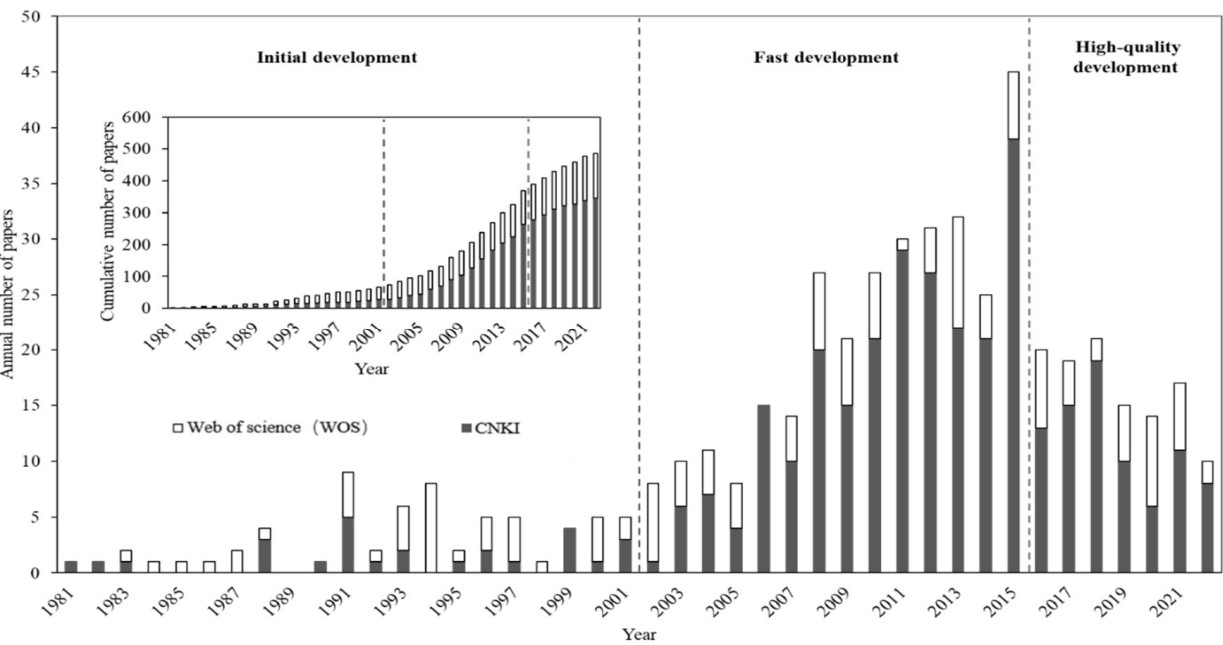

**Figure 4.** Changes in the number of publications of Nepalese alder since 1981.

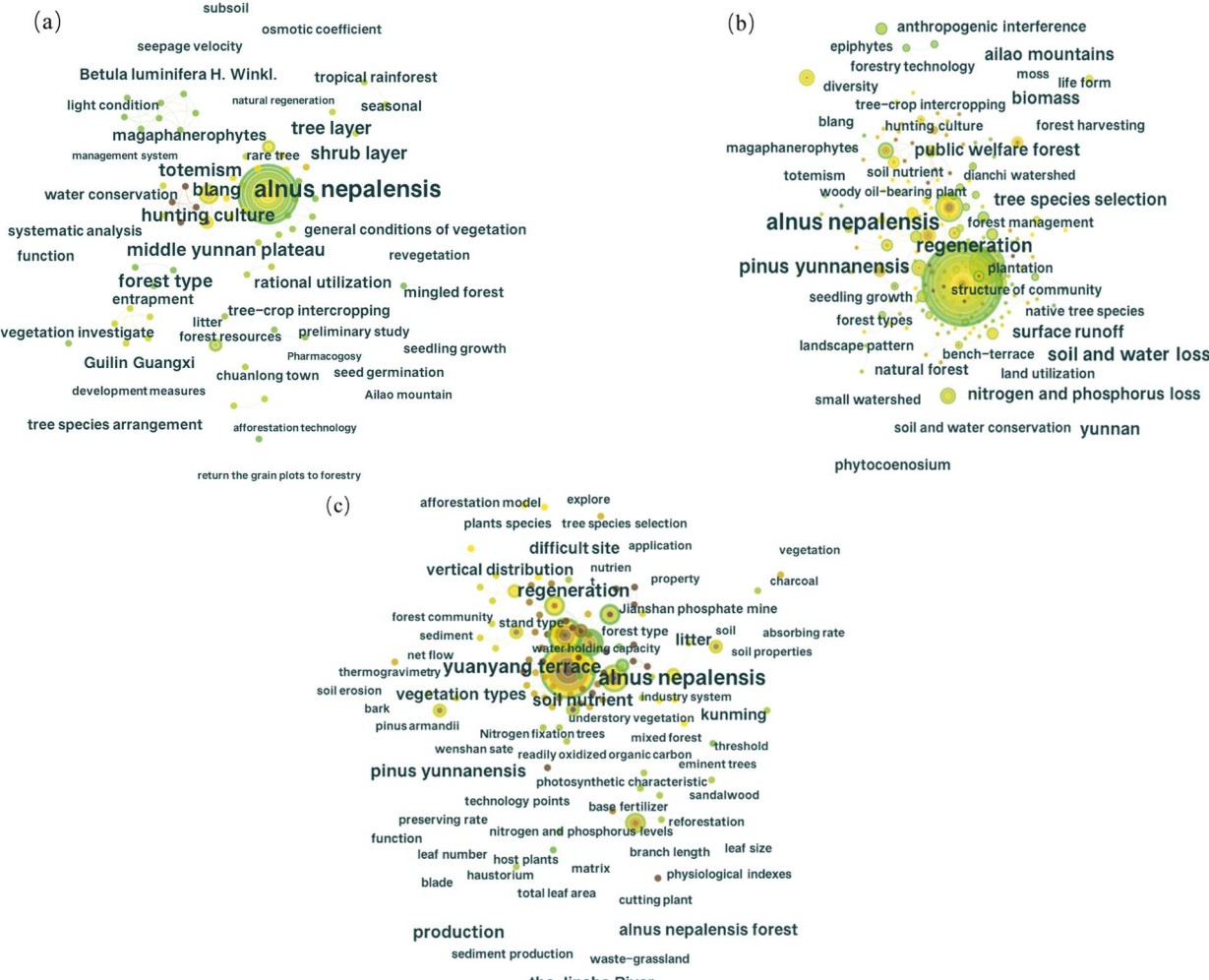

**Figure 5.** Keywords network map for Nepalese alder in different development stages. (**a**) Initial development, (**b**) fast development, (**c**) high-quality development.

**5. Growth Limiting Factors and Sustainable Utilization of Nepalese Alder**

*5.1. Growth Limiting Factors of Nepalese Alder*

The evident northern boundary and upper limit of Nepalese alder suggested that the thermal condition should be the dominant limiting factor. Noshiro et al. studied the effects of tree height, diameter at breast height, and altitude on the anatomical characteristics of Nepalese alder wood in the eastern Himalayas and found that the pore diameter, vessel length, and fiber pipe length of wood were negatively correlated with altitude, indicating that temperature was the main factor affecting its anatomical structure [25]. However, further research by Noshiro et al. suggested that at altitudes above 1800 m, the vessel and fiber lengths of Nepalese alder exhibited a decreasing trend with altitude that may be related to moisture constraints [15]. Quantifying the relative contributions of different hydrothermal factors to its growth and distribution will help to understand the biogeographic distribution pattern of this species; moreover, the selection of suitable ecological sources for plantation can provide important genetic resources reserved for mountain vegetation restoration.

In addition to hydrothermal factors, nitrogen and phosphorus are also important factors affecting the growth of Nepalese alder. Simulated nitrogen deposition experiments showed that the nitrogen addition at lower levels significantly promoted the growth of alder seedlings, whereas that at higher levels inhibited seedling biomass and reduced the investment to allocation of photosynthetic organs [13]. Phosphorus deficiency also decreased chlorophyll content in Nepalese alder seedlings [14]. In addition, environmental factors could indirectly influence the growth of Nepalese alder by affecting the activity or diversity of Frankia. For example, Jha et al. found that phosphate fertilizer treatment reduced mycorrhizal infection in alder seedlings, but it significantly stimulated the nodulation of alder seedlings [26]. It can be seen that phosphorus plays a very important role in the growth of Nepalese alder itself and Frankia, especially in humid tropical and subtropical areas where forest growth is largely affected by phosphorus restriction [27]. In all, the influence of climate, soil, and other factors on the growth and distribution of Nepalese alder should be comprehensively considered in the future.

*5.2. Physio–Ecological Characteristics and Ecological Functions of Nepalese Alder*

As a pioneer species, Nepalese alder often occupies barren hillsides due to their strong adaptability, rapid growth, and strong tolerance to barren soil, and the nitrogen fixation effect of the symbiotic root with Frankia improves soil chemical status to a certain extent. Therefore, by improving the growth environment, Nepalese alder further promotes the growth and development of other plants. At present, most scholars have discussed the physiological and ecological characteristics of Nepalese alder (Figure 6). For example, some studies have explored the growth characteristics of the Nepalese alder and cardamom agroforestry system [11], and the productivity of mixed Nepalese alder and tea plants in gardens [9]. In addition, studies have also found that chemicals in Nepalese alder might affect the growth and development of other plants through allelopathy [28]. The study of Frankia has been a major focus of alder-related research (Figure 6). At this stage, most research has focused on genetic diversity. Studies have shown that Frankia are rich in genetic diversity, which is influenced by various factors such as climate, topography, and altitude. Through the study of the diversity of Frankia, it not only helps people to understand the origin and evolution process of Frankia, so as to build an efficient symbiotic system with strong nitrogen fixation ability and stress resistance, but also provides a scientific basis for revealing the mechanisms of the nitrogen fixation and soil improvement [29].

5.2.1. Soil Improvement

The roots of Nepalese alder attached with Frankia generally form symbiosis nodulation with nitrogen fixation functions [6], thereby changing the soil physicochemical properties and increasing the soil organic matter and nitrogen and phosphorus content (Figure 6). Mishra et al. (2018) analyzed the soil fertility in different forests in the eastern Himalayas, and found that the soil organic matter content of Nepalese alder forests was significantly

higher than that of other forest types, and the presence of Nepalese alder accelerated soil nutrient cycling [8]. Other studies also found that the soil total nitrogen and phosphorus, alkali-hydrolyzed nitrogen, available phosphorus, and soil microbial biomass in Nepalese alder forests were significantly higher than in other vegetation types (e.g., forests dominated by *Pseudognaphalium affine*, *Cunninghamia lanceolata*, *Michelia oblonga*, *Parkia roxburghii*, and *Pinus kesiya*) in the Yuanyang Terraces of Yunnan Province [30] and the hilly ecosystems of Northeast India [31]. Studies have also pointed out that Frankia in the roots of Nepalese alder can not only improve soil physical–chemical properties and increase soil-nutrient content, but also facilitate the reproduction of soil microorganisms and improve the activity of soil enzymes [9]. Li et al. further found that the presence of Nepalese alder increased the number of soil nitrogen-fixing bacteria in seven forest communities (*Clerodendrum bangei*, *Cunninghamia lanceolata*, *Camellia sinensis+Alnus nepalensis*, *Alnus nepalensis*, *Gnaphalium affine*, *Choerospondias axillaris*, and *Neolitzea chui+Schima superba*) in Yuanyang Terrace [32]. Among six different vegetation types in Yuanyang terraces, it was also found that Nepalese alder significantly increased the activity of soil proteases [30].

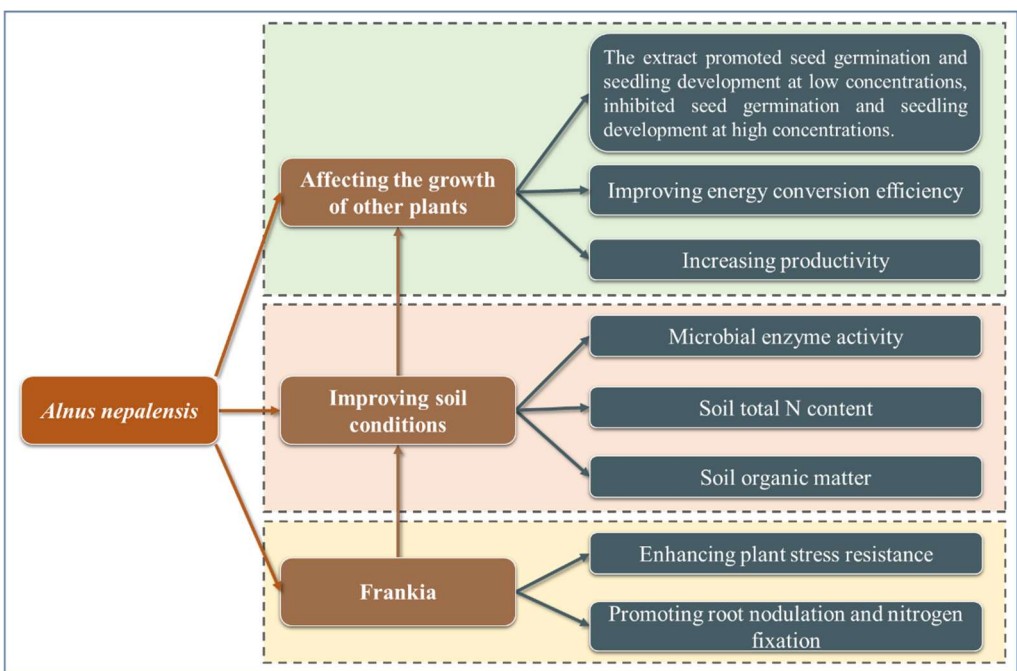

**Figure 6.** A framework for physio-ecological characteristics and ecological functions of Nepalese alder.

5.2.2. Effect of Nepalese Alder on Other Plants

The presence of Nepalese alder can improve soil physical and chemical properties, which in turn promotes the growth of surrounding plants (Figure 6). For example, Mortimer et al. transplanted Nepalese alder to tea (*Camellia sinensis* var. *assamica*) garden, and found that the existence of Nepalese alder increased the variety and number of fungi and bacteria in the soil, which in turn improved the productivity of tea [9]. Sharma et al. found that Nepalese alder significantly increased the soil nitrogen content in the mixed agroforestry system, thereby improving the productivity and energy conversion efficiency of cardamom communities [10]. In addition, the allelopathy of Nepalese alder can also exert a positive or negative impact on the growth of other plants to some extent. Wang et al. studied the effect of fresh leaf aqueous extract of Nepalese alder on the growth of Yunnan pine seedlings and found that the lower concentration of aqueous extract (<5 g/L) can promote the growth of seedlings, enhance root vitality, and increase chlorophyll content [28]. However, a higher concentration above 5 g/L will inhibit the growth of seedlings. Wang et al. further treated the seeds of *P. Yunnanesis* with different concentrations of extracts (800, 400, 150 mg/kg) from different organs of Nepalese alder, and found that the high concentration extracts

significantly inhibited the germination and seedling growth of *P. yunnanensis* [33]. Besides, this inhibition was weakened with the decrease of extract concentration, and eventually turned into a promoting effect. In summary, the effect of Nepalese alder on the growth of other plants is mainly manifested as a promoting effect, while the allelopathy on other plants is a bit more complicated. Generally speaking, a lower concentration of aqueous extract yields a promotion effect, and a higher one may lead to inhibition. That is, when the allelopathic intensity of Nepalese alder itself is high, it will have a negative impact on the growth of other plants, which may explain why Nepalese alder forms a pure forest in the juvenile stage with few other coexisting tree species.

### 5.2.3. Frankia Infection

Frankia infection is the most representative physiological and ecological characteristic of Nepalese alder, and Frankia can form nitrogen-fixing nodules in symbiosis with Nepalese alder roots, which improves the soil conditions in the alder forest to a certain extent (Figure 6). At present, research on Frankia with Nepalese alder has focused on its genetic diversity, nutrient absorption, and morphological variation. Among them, the genetic diversity of Frankia has attracted a lot of attention. Studies have shown that the genetic diversity of Frankia nodules is influenced by a variety of factors, including topography, climate, and altitude [29]. On one hand, Frankia is widely distributed in multiple species, indicating that it can survive in diverse habitats [34]. On the other hand, in different habitats, especially at high altitudes where the environment is harsh, environmental factors such as stronger ultraviolet radiation and drought may cause genetic instability, resulting in more replication errors and higher genetic diversity to meet survival needs [35,36]. Xiong et al. studied the genetic diversity of Frankia in five regions of Yunnan, and found that the distribution and genetic structure of Frankia were closely related to the environment, and there were dominant genotypes in different regions [37]. Tang et al. used rep-PCR to study the genetic diversity of Frankia in Nepalese alder nodules under different habitats in Yunnan, and found the genetic diversity was positively correlated with the degree of environmental stress [29]. Similarly, Dai et al. [34] found that the genetic diversity of the Frankia strain in samples of Nepalese alder nodules in the Hengduan Mountains is related to climate and glacial history. In addition to topographical and climatic factors, altitude is also an important factor affecting the genetic diversity of Frankia, and it is assumed that the greater the altitude gradient, the richer the genetic diversity of Frankia [38,39].

## 6. Perspectives

Although there exist uncertainties about the geographical origin of *Alnus* and the role of climatic factors in shaping its distribution, the above statement indicates that low temperature should be one of the key limiting factors for its distribution. Therefore, in the context of future climate change, Nepalese alder is assumed to dominate in higher latitudes and/or altitudes, which is consistent with the hypothesis that the distribution range of Nepalese alder will expand significantly under the background of climate change, as proposed by Ranjitkar et al. [40]. Based on the maximum entropy model and related climate, soil, and topographic data, the future distribution trend of Nepalese alder under different scenarios can be further simulated.

In addition to the influence of climatic factors, the loss of phosphorus will have an impact on the coupling of Nepalese alder and Frankia to a certain extent [26]. However, there are relatively few studies concentrated on the impact of multi-environmental factors on the growth of Nepalese alder at this stage, which limits our in-depth understanding of the change trend and distribution range of Nepalese alder in future situations. In addition, the Himalayan mountains are very young and still in an active stage. Affected by geology, topography, and climatic factors, as well as the uplift of the Himalayas, the mountains are prone to geological disasters such as landslides and debris flows, which have recently caused a series of ecological and environmental problems such as forest degradation and species habitat destruction [41,42]. Therefore, strengthening the study of environmental

impacts on the growth of Nepalese alder is of great necessity to make full use of Nepalese alder which may improve soil chemical and physical status in early succession, and to accelerate the positive evolution of ecosystem functions.

Previous studies have mentioned the effects of temperature and precipitation on the anatomical structure (i.e., stem traits) of Nepalese alder. Whereas, there is still a lack of research on plant functional traits, especially root and leaf traits, which limits our understanding of the response of Nepalese alder to the changes of hydrothermal factors from different sources. In particular, this species is mainly distributed in tropical and subtropical regions, and the phenomenon of leaf feeding by insects is obvious, and the analysis of leaf morphology and related traits is significant for understanding the plant–insect interaction and its mechanism in light of climate change. This work needs to be carried out urgently.

At this stage, most of the research on Frankia focused on its genetic diversity, especially the environmental impacts (terrain, climate, and altitude) on the genetic diversity, while research on the growth and reproduction of Frankia itself has rarely been reported. Relevant research is important to explore the interconnection between Frankia and Nepalese alder, and to reveal the mechanism affecting the distribution and growth of Nepalese alder. The relationship between Frankia and alder is the key factor in using Nepalese alder for ecological restoration, and it has important theoretical and practical significance for practicing General Secretary Xi Jinping's thesis that lucid waters and lush mountains are golden mountains, and the secret to establishing an ecological civilization highland on the Qinghai-Tibet Plateau.

**Author Contributions:** Conceptualization, L.Z. and C.X.; methodology, L.Z. and C.X.; software, C.X. and W.Z.; validation, L.Z.; formal analysis, C.X.; investigation, L.Z. and C.X.; resources, L.Z.; data curation, C.X. and W.Z.; writing—original draft preparation, C.X.; writing—review and editing, C.X., L.Z., J.W., J.S. and G.C.; visualization, L.Z.; supervision, L.Z.; project administration, L.Z.; funding acquisition, L.Z. All authors have read and agreed to the published version of the manuscript.

**Funding:** The Second Tibetan Plateau Scientific Expedition and Research (STEP) Program (2019QZKK0301-1) and the Key technology research and development projects in Xizang Autonomous Regions (XZ202101ZY0005G) provided financial support.

**Institutional Review Board Statement:** Not applicable.

**Data Availability Statement:** All data used in the manuscript are already publicly accessible, and we provided the download address in the manuscript.

**Conflicts of Interest:** The authors declare no conflict of interest.

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
