# Peer review of "Progress on Geographical Distribution, Driving Factors and Ecological Functions of Nepalese Alder"

_diversity, doi:10.3390/d15010059_

Round 1
Reviewer 1 Report
Dear authors, I found your manuscript is interesting and informative for the Special Issue. The manuscript is well written, clear, and brings to the reader’s attention important topical questions. I have mostly minor comments and suggestions that you may find below.
Minor comments:
Line 3. Different font for “and”.
Line 37. Missing space between "Fig." and "1"
Line 81: Different font for “1” and letter “L”. Also, different fonts used in the manuscript (Lines 110, 171, 192, 225, etc).
Line 126: Did the authors mean critical factors?
Line 161: What aspect did the authors have in mind? Slope?
Figure 4 is not mentioned in the manuscript. Some words are underlined in red by the text editor. Please explain what circles mean in description of the figure.
Line 403: Extra space in “Forestry and Grassland”
Author Response
We thank the reviewer for the valuable comments and suggestions. In this version, we have deleted Figure 4 and the supplementary Table S1, and their corresponding text to let the story more readable and focused. We have also revised the manuscript in response to all the comments and suggestions. Detailed responses are as follows.
Question 1 Line 126: Did the authors mean critical factors?
Reply:Sorry for the wrong typewriting, we have revised it as climatic.
Question 2 Line 161: What aspect did the authors have in mind? Slope?
Reply:Thanks for your carefulness. Slope aspect was analyzed here, and the corresponding parts have been revised. Please go to line 163 in the revised version.
Question 3 Figure 4 is not mentioned in the manuscript. Some words are underlined in red by the text editor. Please explain what circles mean in description of the figure.
Reply:In this version, the original Figure 4 has been deleted since this part seems redundant.

Reviewer 2 Report
According to my opinion, the topic is really actual in the time of ongoing climate change. But also I think that the work need some revisions. Mainly, I do not understand the importance of Bibliometrics analysis and it seems as something redundant which do not enrich the paper. If you want to keep it here, it is necessary to better describe these analyses and results and to better connect it with the rest of paper. Remaining chapters are nice interesting review for me and I have only some minor comments to them.
line 126? clitical??? should it be climatic???
line 110: chapter 2.3 Bibliometric analysis, I don´t know this technic and from your description it is not clear for me what does it mean. Please, decribe this method more deeply in order to also properly understand results in chapter 4.1 which are also not clear for me.
page 6, Figure 4: I do not understand the values mentioned in the diagram. I prefer to mention log-likelihood ratio values here.
line 193: Was there some criteria for delimitating these categories?
e.g. line 200 and 208: It is weird for me not to mention publication year after the name of author.
line 274: I would appreciate better interconnection between Figure 7 and the text.
lines 282-289: I think it is not necessary to mention nearly the same examples in three sentences. Please, condense it.
line 299: use Alnus nepalensis or Nepalese alder
Author Response
We thank the reviewer for the valuable comments and suggestions. In this version, we have deleted Figure 4 and the supplementary Table S1, and their corresponding text to let the story more readable and focused. We have also revised the manuscript in response to all the comments and suggestions. Detailed responses are as follows.
Question 1 According to my opinion, the topic is really actual in the time of ongoing climate change. But also I think that the work needs some revisions. Mainly, I do not understand the importance of Bibliometrics analysis and it seems something redundant which do not enrich the paper. If you want to keep it here, it is necessary to better describe these analyses and results and to better connect them with the rest of the paper. The remaining chapters are nice and interesting review for me and I have only some minor comments on them.
Reply:Since this part is not so closely related to our topic, as well as its obscure expression, we have deleted section 4.1 and the corresponding Figure 4 and Supplementary Table S1. We believe the new version would be more concise.
Question 2 line 126 clitical? should it be climatic?
Reply:Thanks, and we have revised it as climatic.
Question 3 line 110: chapter 2.3 Bibliometric analysis, I don´t know this technic and from your description it is not clear for me what does it mean. Please, decribe this method more deeply in order to also properly understand results in chapter 4.1 which are also not clear for me.
Reply:We have deleted the part of 4.1 to let the whole story more readable.
Question 4 page 6, Figure 4: I do not understand the values mentioned in the diagram. I prefer to mention log-likelihood ratio values here.
Reply:The original Figure 4 has been deleted in the new version.
Question 5 line 193: Was there some criteria for delimitating these categories?
Reply:Please go to page 6 line 204: “In 2002, it entered into an accelerated period, during which the number of publications increased significantly, and reached a peak in 2015 (39 articles), which was about 10 times that of 2005 (the least number of articles published in this period). This might be related to the national policy to fully launch the project of returning farmland to forest in 2002, as well as the implementation of related projects which greatly promoted the research on the fast-growing alder tree species”, and line 217 “After 2015, the number of publications showed a downward trend, which might be related to the fact that the research has focused more on practical applications.”
Question 6 lines 282-289: I think it is not necessary to mention nearly the same examples in three sentences. Please, condense it.
Reply:Thanks for the suggestions. We have condensed this part. Please go to page 9 line 287 “Other studies also found that the soil total nitrogen and phosphorus, alkali-hydrolyzed nitrogen, available phosphorus and soil microbial biomass in Nepalese alder forests were significantly higher than in other vegetation types (e.g., forests dominated by Pseudognaphalium affine, Cunninghamia lanceolata, Michelia oblonga, Parkia roxburghii, and Pinus kesiya) in the Yuanyang Terraces of Yunnan Province [30] and the hilly ecosystems of Northeast India [31].”
Question 7 line 299: use Alnus nepalensis or Nepalese alder
Reply:Thanks for the suggestions, and we have replaced A. nepalensis with Nepalese alder in most cases in the new version.

Round 2
Reviewer 2 Report
After revison, I recommend to accept it in the present form.